# Development and Experimental Analysis of a Seeding Quantity Sensor for the Precision Seeding of Small Seeds

**DOI:** 10.3390/s19235191

**Published:** 2019-11-27

**Authors:** Wei Liu, Jianping Hu, Xingsheng Zhao, Haoran Pan, Imran Ali Lakhiar, Wei Wang, Jun Zhao

**Affiliations:** School of Agricultural Equipment Engineering, Jiangsu University, Zhenjiang 212013, China; lw19900714@163.com (W.L.); zxs19950316@163.com (X.Z.); panujs@163.com (H.P.); 5103160321@stmail.ujs.edu.cn (I.A.L.); 17826077597@163.com (W.W.); ZJ100670@163.com (J.Z.)

**Keywords:** small seed, seeding quantity, seed falling trajectory controlling device, seed-counting algorithm, structure optimization experiment

## Abstract

Having the correct seeding rate for a unit area is vital to crop yields. In order to assess the desirable seeding rate, the number of discharged seeds needs to be monitored in real-time. However, for small seeds, the miscounting of seeds during monitoring happens frequently when using conventional seeding quantity sensors, which have wide light beam intervals. Thus, a seeding quantity sensor, which enables small seeds to pass through the light beam steadily, was developed. Based on the seed-shading time, a seed-counting algorithm was proposed. Moreover, the key structure parameters of the proposed sensor were ascertained using an optimization experiment. Finally, the developed seeding quantity sensor was tested against a photoelectric sensor and a fiber sensor to compare the seed monitoring accuracies. The results show that the average monitoring accuracy of the developed sensor, photoelectric sensor, and fiber sensor were 97.09%, 56.79%, and 91.10%, respectively. Furthermore, the factorial analysis shows that the forward velocity of the experimental apparatus and the rotational speed of the seeding plate did not significantly change the monitoring accuracies obtained by the developed sensor. Therefore, the developed sensor can be applied to monitor the seed quantity for the precision seeding of small seeds accurately and robustly.

## 1. Introduction

Some varieties of vegetable seeds, such as pakchoi, celery cabbage, and so on, are planted using the precision seeding mode, which involves placing seeds one by one with a required distance [1]. This is done because an appropriate amount of seeds at a uniform distribution can enable seeds to get enough water, nutrients, space, and other living necessities [2,3]. In order to assess the seeding rate, the number of discharged seeds need to be monitored in real-time during seeding processes. Nowadays, some seeding quantity sensors have been developed, and the majority of them are based on piezoelectric, capacitance, radio wave, and photoelectric sensing theories.

Concerning the piezoelectric sensor, its core component is a pressure-sensitive element that can transform a seed’s impact into an electric signal. Ding et al. [4] developed a piezoelectric sensor for rapeseeds by using polyvinylidene flouride (PVDF) material. Their results showed that the detection accuracy declined as the rotational speed of the seeding plate increased. Another study by Huang et al. [5] designed a piezoelectric sensor for counting the number of discharged corn seeds. The results showed that the monitoring accuracy was 92.5%. However, some seeds would bounce up after impacting on the pressure-sensitive element. In addition, they may hit the element again, resulting in being counted more than once [6].

As for the capacitance sensor, the dielectric coefficient of the sensor has a linear relationship with the seed mass between the two electrodes [7]. Sun et al. [8] designed three kinds of capacitance sensors, namely the crossed E-shape plate, differential tri-plate, and parallel plate type, and compared their sensitivities in terms of counting corn seeds. This study showed that the E-shape plate and the differential tri-plate capacitance sensor were able to sense corn seeds accurately. Similarly, a study by Zhou et al. [9] developed a capacitance sensor for detecting the number of discharging corn seeds in the precision seeding mode. The results showed that the sensor effectively distinguished a single seed falling alone or two seeds falling together by integrating the area of the signal generated by falling seeds. The capacitance sensing system can detect more than one seed at a time. However, capacitance sensors are easily influenced by the seed’s water content, machine vibrations, temperatures, its own parasitic capacitance, etc. [10]. Moreover, the seeds with a small volume cannot cause a detectable variation of the dielectric coefficient.

By detecting seeds via high-frequency radio waves, the WaveVision sensor (Precision Planting LLC, Tremont, IL, USA) could judge the seed number in the seed tube using the seeds’ mass [11]. In addition, the radio waves can monitor seeds through dust. The WaveVision sensor can only detect big seeds, such as beans, corn, pelleted sugarbeet seeds, and so on, because small seeds cannot reflect radio waves strongly enough to be detectable. Therefore, the WaveVision sensor cannot be used to count the number of small seeds.

A photoelectric sensor often includes a transmitter and a receiver. In this type, if the light beam emitted by the transmitter is interrupted by a seed, the output of the receiver would change. In recent years, several studies had been conducted by developing the photoelectric sensors for monitoring falling seeds [12]. Kostić et al. [13] used photoelectric sensors to monitor the number of falling corn seeds ejected by a seed-metering device. Furthermore, an infrared sensing system was developed for the rapid evaluation of the discharged vegetable seeds in laboratory conditions [14]. It was found that the seeds whose diameter was less than 3 mm may not totally block a light beam and change the output of the sensor. Moreover, Deividson et al. [15] applied a reflective infrared sensor to detect the corn seeds on a grease belt and the results showed that the detection accuracy was more than 99%. A photocell sensor (MC ELECTRONICS Co. Ltd., Fiesso Umbertiano, Italy) was designed for small seeds, such as carrots, lettuce, radishes, etc. It can detect the seeds whose diameter is only 0.3 mm, but it cannot judge the exact number when some seeds are occluded by another seed [16].

In addition, due to some varieties of seeds having a large volume, just a few light beams can monitor the majority of falling seeds [17]. In order to avoid interference, there should be a clearance between any two adjacent light beams [18]. However, a small seed’s diameter is often less than 3 mm, and the trajectory of a falling seed cannot be controlled. Thus, falling seeds with a small volume can easily pass through the clearances and lead to a missing detection [19].

To cover the whole width of a seed falling area without any clearance, some photoelectric sensing systems with a wider monitoring area were developed [1]. Hajamed et al. [20] applied a fiber sensor to monitor the number of chickpea seeds under laboratory conditions. They reported that the overall errors were ±4% and ±10% at the forward speeds of 1.3 m·s^−1^ and 1.9 m·s^−1^, respectively. Another study by Zhang et al. [21] developed a sensor that integrated eight infrared sensing units to detect rapeseed, and the results showed that the accuracy of the sensor was about 90%. Moreover, according to the reported studies, although the monitoring width of the fiber sensor has improved, its monitoring accuracy and robustness are still not acceptable. This might have been caused by the interference of machine vibrations, natural lights, dust, and so on [22].

In terms of conventional photoelectric sensors, there is no seed-counting algorithm reported in the literature. Hence, due to malfunctions of the seeding plate, it is possible that two or more seeds consecutively pass a light beam without a space interval, and the leads to those seeds only being counted once [23]. Furthermore, some tiny particles, such as broken seeds and tiny stones, would be viewed as seeds due to the lack of a seed-counting algorithm. Thus, it is necessary to develop an algorithm for counting the exact number of small seeds.

Thus, this study aimed to develop a seeding quantity sensor, which was based on a new algorithm used for small seeds whose diameter was less than 3 mm. The content of this paper is as follows: First, the structure of the sensor was designed. Furthermore, based on the developed sensor, an algorithm was proposed to count the exact number of falling seeds. The algorithm could distinguish not only the tiny particles but also the overlapping seeds. In the next step, optimization experiments were conducted to obtain the optimal sensor structure with the highest monitoring accuracy. Finally, an accuracy comparison experiment was conducted between the developed sensor and two off-the-shelf sensors.

## 2. Design of a Seeding Quantity Sensor

### 2.1. Working Principle of the Seeding Quantity Sensor

Figure 1 shows the structure of a typical seed-metering unit, which was composed of a hopper, a seeding plate, a seed-protecting device, and a seeding quantity sensor. Furthermore, the seeding quantity sensor consisted of a taper inlet, a seed-guiding slot, an infrared ray transmitter, and an infrared ray receiver. As the seeding plate rotated anticlockwise, the seeds in the hopper would fill the holes on the perimeter of the seeding plate, and then be discharged at the end of the seed-protecting device. In order to monitor the number of discharged seeds, a seeding quantity sensor need to be installed at the outlet of the seed-metering device.

Conventional opposite-type seeding quantity sensors covered the whole width of the seed-falling area by using several parallel lights. Nevertheless, if light beams were arranged too narrow, adjacent light beam may interfere with the receiver. In contrast, if there was a space interval between any two light beams, some small seeds may pass through the space interval without being measured [24]. In order to avoid interference, only an opposite-type infrared ray sensing unit was applied in this study. When a falling seed interrupted the infrared ray, the voltage of the infrared ray receiver would change. Thus, according to the voltage of the receiver, whether there was a seed falling through the seeding quantity sensor could be judged.

### 2.2. Design of the Falling Seed Trajectory Controlling (FSTC) Device

The FSTC device consisted of a seed-guiding slot and a taper inlet. The seed-guiding slot was designed to enable the falling seeds to pass through the infrared ray steadily. Moreover, the taper inlet can help seeds enter the seed-guiding slot smoothly.

As seen in Figure 2, a coordinate system was set up on the center of a seed plate. When seed was discharged by the seeding plate, its theoretical trajectory was a parabola. In this research, the height between the seed-discharging point and the entrance of the seed-guiding slot was 41 mm. The formula for the trajectory in X- and Y-directions were:(1)x=ω·r·t
and
(2)$$y=\frac{1}{2}\cdot g\cdot t^{2}$$
where *x* indicates the seed falling displacement in the X-direction (m);

*ω* is the angular velocity of the seeding plate (rad·s^−1^); in this research, the range of *ω* was between 0.52 and 2.10 rad·s^−1^;

*r* represents the radius of the seeding plate (m); in this study, it was 0.03 m;

*t* represents the time the seed was falling (s);

*y* is the falling seed’s displacement in the Y-direction (m);

*g* indicates the acceleration due to gravity (9.81 m·s^−1^);

Substituting the value of *y* into Equation (2), a seed’s falling time from when it was discharged from the seeding plate until falling into the guiding slot was determined, which was approximately 0.09 s. Moreover, the installation distance in the X-direction between the middle point of the seed guiding slot and discharging point was 5.67 mm, which was calculated using Equation (1). To avoid a choking phenomenon, the length of guiding slot *L* should be larger than the largest diameter (2.14 mm) of the seeds. However, if the distance between the transmitter and receiver was more than 5.5 mm, the receiver may not receive the infrared ray steadily. Therefore, the range of *L* was determined to be between 3 to 5.5 mm.

Concerning the inclination angle *α*, if it was more than 60°, the inlet would become small and could not cover the whole outlet of the seed-metering device. However, if the inclination was less than 30°, the sensor would be bulky and difficult to be integrated onto the seed-metering device. Therefore, the inclination angle *α* of the tapered wall was between 30°and 60°.

As for the width of the seed-guiding slot, it also needed to be designed. The cross-section diagram of the guiding slot is shown in Figure 3. The diameter of the infrared ray must be less than the seeds’ minimum diameter. Otherwise, it was possible that more than one seed shaded the infrared ray at a time. Thus, the diameter of the infrared ray was set to be 1.3 mm. In addition, the width of the guiding slot *W* must meet the following condition:(3)$$D_{\max}<W<2.5\cdot D_{\min}$$
where *D*_max_ and *D*_min_ represent the maximum and the minimum of the seeds’ diameters (mm); they were 2.14 and 1.36 mm, respectively.

Furthermore, the *W* must be larger than the maximum of the seed diameter *D*_max_ (Figure 3a). Otherwise, the seeds whose diameters are larger than *W* would get stuck at the entrance of the seed-guiding slot. In the other situation, even if the smallest seeds fall along the edge of the guiding slot (Figure 3b), at least two-fifths of the infrared ray must be shaded. If not, the output of the receiver will not be triggered steadily.

### 2.3. Signal Processing Circuit

The opposite-type infrared ray sensing unit (Beike Shangmao Co. Ltd., Shenzhen, China), used to detect small seeds, consisted of a transmitter and a receiver [25,26,27]. The receiver was installed such that it pointed toward the transmitter to ensure that it received the infrared ray emitted by the transmitter.

The schematic diagram of the signal processing circuit is shown in Figure 4. If the infrared ray was interrupted by seeds, the voltage at the IN+ port would exceed the reference voltage at IN−. Thus, the comparator LM393 would output a high-level pulse. In contrast, if the voltage at the IN+ was less than the reference, the output of the LM393 would maintain a low level. Nevertheless, in the practical test, there were some high-frequency noises at two ends of the output signal. The main reason might be that the occluded area was not steady when a seed entered and left the infrared ray [28]. In order to acquire a more accurate detection signal, a capacitance C0 was used for filtering the high-frequency noises. Moreover, the output pin was connected with a micro control unit (MCU).

## 3. Materials and Methods

### 3.1. Seed-Counting Algorithm (SCA)

As for traditional seeding quantity methods, the total seed number would increase by one when the MCU detected an interrupted signal [29,30]. Those methods, therefore, cannot distinguish between sand and broken seeds during the monitoring process because sometimes these tiny particles can also trigger interrupted signals. In addition, if two or more seeds consecutively fell through the light hole without any space interval, they would be viewed as one seed by mistake. Based on the single seed falling time, a seed-counting algorithm (SCA) was proposed to overcome the aforementioned drawbacks. Figure 5 displays the flow chart of the SCA.

As previously mentioned in Section 2.3, the signal processing circuit would output a high-level impulse when the infrared ray was shaded by a falling seed. If the monitoring pin (MP) of the MCU detected a rising edge, a pulse-width timer would start to count the sustaining time *T_sustain_* of the high-level. The *T_sustain_* would continuously increase until the MP detected a falling edge.

In the next step, the *T_sustain_* was compared to the minimum time interval *T_low_* and the maximum time interval *T_high_.* If the *T_sustain_* was less than the *T_low_*, it inferred that a small particle, such as sand, a broken seed, and so on, had passed through the infrared ray. Thus, the total number *N* would maintain the same value.

If the *T_sustain_* was greater than the *T_low_* but still lower than the *T_high_*, the total number of the fallen seeds *N* would increase by one. According to observations, it was impossible that three or more seeds were falling together. Hence, when the *T_sustain_* exceeded the *T_high_*, it revealed that two seeds fell consecutively without any space interval. Also, the total number *N* would add two at a time. After the seed-counting process, the MP was reset and waited for the next rising edge. The judging procedure can be illustrated using Equation (4):(4)$$\left\{ \begin{array}{cc} N=N & T_{sustain}\;<\;T_{low}\\ N=N+1 & T_{low}\;<\;T_{sustain}\;<\;T_{high}\\ N=N+2 & T_{high}\;<\;T_{sustain} \end{array}\right.$$

### 3.2. Experimental Apparatus

An experimental apparatus (Figure 6) for seeding performance monitoring was applied. It could conduct seeding performance monitoring experiments in a mobile state. The movable experimental apparatus was composed of a hopper, a controlling box, a seed-metering device, two stepper motors, four driving wheels, and aluminum alloy frames. In addition, a human–machine interface (HMI) (Figure 7) was developed as an application software installed on a tablet computer (Ipad5, Apple Inc., Cupertino, CA, USA).

Furthermore, the layout of the hardware structure of the experimental apparatus is shown in Figure 8. The control system consisted of two sections: a slave detection system and the main detection system. Furthermore, the slave detection system was used to detect the output signals generated by the signal processing circuit, while the main detection system, the upper computer (cMT-Server-100, WEINVIEW Co. Ltd., Shenzhen, China), was used to read seed numbers from the slave detection system.

Further, a programmable logical controller (PLC) (FX-1N, Mitsubishi Electric Corporation, Tokyo, Japan) connected to the upper computer was applied to control the forward speed of the experimental apparatus and the rotational speed of the seeding plate. Furthermore, a router (TL-WDR5620, TP-Link Technologies Co., Ltd., Shenzhen, China) was applied to transmit the detection information to the HMI using a wireless signal and receive the commands from the HMI to control the PLC. In addition, a container was placed under the sensors to catch the discharged seeds. After each experiment, the actual seed number was counted manually.

### 3.3. Experimental Analysis

#### 3.3.1. Determining the Time Thresholds of a Common Seed

In order to determine the values of *T_high_* and *T_low_*, the falling tests were conducted using three types of seeds, namely common-sized seeds, broken seeds, and double seeds. Furthermore, the broken seeds were made by cutting a common sized seed into two parts, and two common-sized seeds were stuck together to simulate two seeds falling without space. Each mentioned condition was repeated a hundred times. The vegetable seed variety used in this study was the pakchoi seed, whose shape was spherical, and the average diameter was 1.71 mm.

Furthermore, because the time thresholds determination experiment was carried out before the structure optimization experiment, the structure parameters of the FSTC device were set as the median of the value ranges. That is, the length and width of the guiding slot, as well as the inclination angle of the inlet, were 4.25 mm, 2.75 mm, and 45°, respectively.

#### 3.3.2. Optimization Experiment for the Structure of the FSTC Device

In order to determine the optimal structure parameter of the FSTC device, an optimization experiment was conducted, which was based on the response surface method (RSM). According to the analysis in Section 2.2, the inclination angle of the inlet, as well as the length and width of the seed-guiding slot, may have effects on the seed number monitoring. Therefore, they were selected as factors for the structure optimization experiment. However, Table 1 presents the level values of these factors.

The optimization experiment was designed using the Design-Expert software (10.0.7, Stat-Ease Inc., Minneapolis, MN, USA). There were 20 experiments in total, and the center point experiment was repeated six times. The experimental apparatus traveled 10 m during each experimental time. The forward velocity of the experimental apparatus and rotational speed of the seeding plate were 1 m·s^−1^ and 10 rev·min^−1^, respectively. The response of the experiment was the monitoring accuracy of the developed sensor. Moreover, the monitoring accuracy was calculated as follows:(5)$$Accuracy=\left(1-\left|\frac{V_{m}}{V_{a}}-1\right|\right)\times 100\%$$
where *V_a_* was the actual seed number in an experiment; *V_m_* was the seed number monitored by the developed sensor. When an experimental FSTC device obtained the highest accuracy, its factor levels were selected as the optimal structure parameters.

#### 3.3.3. Accuracy Comparison Experiment

A micro photoelectric sensor and a fiber sensor were used for the comparison with the developed one in terms of seed number monitoring. The micro photoelectric sensor (PRM51-N1, KEYENCE Corp, Osaka, Japan) was composed of a transmitter and a receiver. When the light beam was interrupted by seeds, the receiver would output a low level. In contrast, if there was nothing between the transmitter and the receiver, the photoelectric sensor would maintain a high-level output.

The fiber sensor consisted of a transmitter, a receiver, and an amplifier (FU-A40 and FS-N11MN, KEYENCE Corp, Osaka, Japan) [23]. There were forty light beams emitted by the transmitter to the receiver. Furthermore, the distance between the transmitter and receiver was 40 mm.

The fiber sensor needed to be calibrated before being used. First, the light intensity of the fiber sensor was set to the “Fine” mode, and the light saturation function needed to be turned off. In the calibrating process, several pakchoi seeds needed to be selected randomly and then discharged between the transmitter and the receiver. Using the full auto preset function, when the biggest and the small seeds interrupted the light beams, the light intensity value was calibrated to 0 and 100, respectively. Afterward, if there was a seed shading the light beams, the light intensity would reduce to a value between 0 to 100, which was proportional to the light area shaded by the falling seed. Moreover, a trigger threshold needed to be set. Once the light intensity value was less than the threshold, the fiber sensor would generate a low-level impulse. In order to balance the sensor sensitivity to the seeds and apparatus vibrations, the threshold was set as 95. When the MCU detected a low-level impulse, it would consider that there was a seed passing through the fiber sensor. Furthermore, the fiber sensor should be fixed on a “∏”-shaped holder under the developed sensor, as seen in Figure 6.

In the practical seeding process, both the forward speeds of a seeder and rotational speeds of a seeding plate played vital roles in the seeding performance. A completely randomized experiment was used to compare the accuracy of the three sensors under three forward velocities and three rotational speeds of the seeding plate. In practice, the value range of the forward speed of a vegetable precision seeder was between 0.5 and 1.3 m·s^−1^, and that of the rotational speed of the seeding plate and was between 5 and 20 rev·min^−1^. The level values of forwarding velocity and rotational speed of the seeding plate are listed in Table 2. In each treatment, the experimental apparatus traveled 20 m on a rough cement floor.

### 3.4. Statistical Calculation Methodology

The factor analysis at a 0.05 level was considered the significant value to determine whether the forward velocity of the experimental apparatus and rotational speed of the seeding plate could significantly affect the monitoring accuracy of the developed sensor. A commercial data processing software IBM SPSS Statistics (Version 21.0.0.0, International Business Machine Corp.,Armonk, NY, USA) was used to perform the factor analysis in this study.

## 4. Results and Discussion

### 4.1. Result Analyses of the Seed-Shading Time Thresholds Determination Experiment

The frequencies of the seed-shading time under three experimental conditions are shown in Figure 9. In this figure, the frequency indicates the number of times a seed-shading time occurred under each seed volume condition (e.g., the number 47 denotes there were 47 times when the seed-shading-time was in between 1 to 2 μs under the broken seed falling condition). According to the results in Figure 9, 88% of the broken seeds’ shading times was less than 4 μs, and 86% of the double seeds shaded the infrared ray for more than 6 μs; thus, the *T_low_* and *T_high_* were set to 4 and 6 μs, respectively.

As seen in Figure 9, there was still a little shading time caused by broken seeds of more than 4 μs; this may have been caused by seeds’ different falling postures. If the diameter of a broken seed was vertical to the infrared ray when it fell through the seed guiding slot, the shading time would be the same as that of a common seed, which can be seen in Figure 10a. However, in most cases, the angle between the seed’s diameter and the infrared ray was less than 90°, such that most of the broken seed-shading times were less than 4 μs (Figure 10b). Also, due to a similar reason, the shading time of stuck seeds may be less than 6 μs.

As for common seeds, a part of the seed-shading time may exceed the *T_high_* since some common seeds may repeatedly bounce in the seed-guiding slot, as seen in Figure 11. Such a bouncing phenomenon might increase the seed-shading time when they pass through the infrared ray. If a seed passed through an infrared ray without any bounce, it would pass the infrared ray quickly because the total journey was just equal to the sum of the diameter of the infrared ray and that of the falling seed. However, owing to bounces, a seed may pass the infrared ray with a longer journey. Hence, the seed-shading time would be more than 6 μs. According to the SCA, the total seed number would add two rather than one.

### 4.2. Results of the Structure Optimization Experiment for the FSTC Device

The structure optimization experiment for the FSTC device was designed using the RSM. Detailed level combinations and their average monitoring accuracies are shown in Table 3. The SCA algorithm was used for judging the falling seed number in the optimization experiment. According to the results, the recommended slot length, slot width, and inlet angle were 5.50 mm, 3.00 mm, and 30.00°, respectively.

The developed seeding quantity sensor with the optimal structure parameters was validated under the same experimental conditions in triplicate (i.e., forward speed and rotational speed were still 1 m·s^−1^ and 10 rev·min^−1^). The monitoring accuracies obtained in the validating experiments were 97.37%, 99.32%, and 100%. The average accuracy of the validating tests was 98.90%; thus, this combination of levels could be viewed as the optimal structure parameters.

### 4.3. Accuracy Comparison between the Three Sensors

Figure 12 shows the average accuracies detected by the three sensors for all experimental levels. If the total accuracies in all experiment levels were viewed as a whole data set, the average accuracies of the three sensors were calculated and are shown by the last group of bars.

In Figure 12, all of the average accuracies monitored by the PRM51-N1 were lower than 63%. In addition, in all experimental levels, the detection values obtained by the PRM51-N1 were lower than the actual values, which indicates that missing detections frequently happened when the PRM51-N1 was used for monitoring the seed number. There are two reasons that may cause such a phenomenon:The light beam of the PRM51-N1 could not cover the whole seed-falling area. That is, not all falling seeds could interrupt the light beam steadily. Hence, some seeds may have missed being monitored during the seeding process.The light beam emitted by the photoelectric transmitter was wider than the diameters of some seeds such that the seeds could not shade enough light to trigger the PRM51-N1 to produce a low-level impulse signal.

According to the monitoring accuracies in the above analyses, we can conclude that the photoelectric sensor PRM51-N1 could not be used for monitoring seeding performances in practice.

As for the developed sensor and the fiber sensor FU-A40, the monitoring accuracies obtained by the developed sensor were higher than those of the FU-A40 in seven comparison tests. Only at two experimental levels (V3R1 and V3R2) were the monitoring accuracies of the fiber sensor equal to those of the developed sensor. This indicates that the developed seeding quantity sensor had a better monitoring performance versus the fiber sensor, except when the apparatus ran at 1.5 m·s^−1^.

According to the accuracy results of the developed seeding quantity sensor, the reasons for monitoring errors can be explained as follows:As seen in Figure 12, with the same rotational speed, the monitoring accuracies tested by the developed sensor gradually dropped off as the forward velocity rose, but the trend was not strongly apparent. It indicated that the forward speed might slightly affect the monitoring accuracy of the developed sensor.A part of monitoring errors may have been caused by the rough floor. The vibrations caused by the bumpy landform may have let falling seeds repeatedly bounce in the seed-guiding slot (Figure 11), thereby prolonging the seed-shading time and reducing the monitoring accuracy.The seed’s falling posture may have also affected the seed shading time and led to a miscounting of the seed number, which has been illustrated in Section 4.1.

Additionally, there were three reasons to explain why the monitoring performance of the developed sensor was better than that of the fiber sensor:
Owing to the vibrations of the experimental apparatus, the fiber transmitter and the fiber receiver may not have totally aligned, which resulted in the light intensity value being less than the set threshold (95). Sometimes, the monitoring value of the fiber sensor would still increase one or two when the experimental apparatus stopped because the experimental apparatus would vibrate back and forth several times after rapidly braking.As the time of use increased, the temperature of the fiber sensor would rise, which led to the default light intensity value rising with it. Thus, with a long usage time, the sensitivity of the fiber sensor would reduce gradually because it was difficult for falling seeds to shade enough light area to make the light intensity value be less than the threshold. That is why the monitoring accuracy of the fiber sensor in the last two experiments (V2R1 and V2R2, the performing sequence of experiments are shown in Figure 12) were relatively low. In contrast, the developed sensor was stable and robust, no matter how long it was used, because it was not sensitive to the temperature of inner resistances. We can conclude that the monitoring performance of the developed sensor was steady because all of its monitoring accuracies were higher than 91%.In terms of the fiber sensor, the volumes of the seeds used in the calibration played a pivotal role in its monitoring performance. If a falling seed was smaller than the smallest seed used in the calibration, it could not trigger the fiber sensor to generate a low-level impulse; therefore, it could not be monitored. However, the developed sensor did not need to be calibrated because the SCA algorithm could judge seeds automatically. Hence, the developed sensor could not only save the time otherwise used for calibrations, but could also monitor falling seeds without calibration errors.

### 4.4. Statistical Analysis

Under the experimental condition, to determine whether the forward velocity of the experimental apparatus and rotational speed of the seeding plate significantly affected the monitoring accuracy of the developed sensor, a factorial analysis was undertaken. The results of the factorial analysis are shown in Table 4.

According to the results in Table 4, since the significance values of the forward velocity, rotational speed, and their intersection term were higher than 0.05, the null hypothesis (i.e., the factor does not have a significant impact on the monitoring accuracy) was accepted. It revealed that neither the rotational speed nor the forward velocity could significantly affect the monitoring accuracies of the developed sensor. Furthermore, the statistical results also demonstrated that the developed sensor was steady and robust in a practical small-seed counting process.

## 5. Conclusions

In this research, a seeding quantity sensor for small seeds was developed. In order to enable discharged seeds to pass through the infrared ray steadily, an FSTC device was first designed. Moreover, structure optimization experiments were carried out to find the optimal structure parameters of the FSTC device. In addition, a seed-counting algorithm was proposed for distinguishing the tiny particles, common seeds, and overlapping seeds. Finally, accuracy comparison experiments were conducted between the developed seeding quantity sensor and two off-the-shelf sensors under experimental conditions. During the study, several conclusions were drawn by comparing the monitoring accuracies and analyzing the statistical results:According to the results of the structure optimization experiment, the optimal structure parameters of the FSTC device were obtained. In detail, the length of the seed-guiding slot, the width of the seed-guiding slot, and the inclination angle of the inlet were 5.5 mm, 3.00 mm, and 30°, respectively. The developed seeding quantity sensor with the optimized structure was validated three times, and the average accuracy was 98.90%.By comparing the average accuracies obtained by the three sensors for each experiment, the monitoring accuracies of the developed sensor were at least as high as those detected by the other two sensors. Furthermore, if all monitoring accuracies obtained by the same sensor were viewed as a whole data set, the average monitoring accuracy of the developed sensor, photoelectric sensor, and fiber sensor were 97.09%, 56.79%, and 91.10%, respectively. Thus, it is wise to select the developed sensor to monitor the quantities of discharged small seeds for practical seeding processing.According to the factorial analyses, the two factors, namely the forward velocity and the rotational speed of the seeding plate, did not have a significant impact on the monitoring accuracies obtained by the developed seeding quantity sensor. The results demonstrated that the developed seeding quantity sensor is statistically steady and robust under experimental conditions.

In the future, based on the developed sensor, we intend to provide a planting space measurement by recording the time interval between two continuous falling seeds. Moreover, the field experiments would be carried out to validate the performance of the developed sensor under higher seeding frequencies. Furthermore, the developed seeding quantity sensor can also be combined with GPS to generate seeding prescription maps, which can be employed in achieving precision operations for subsequent agronomy processes, such as fertilization, pesticide spraying, and harvesting.

## Figures and Tables

**Figure 1 sensors-19-05191-f001:**
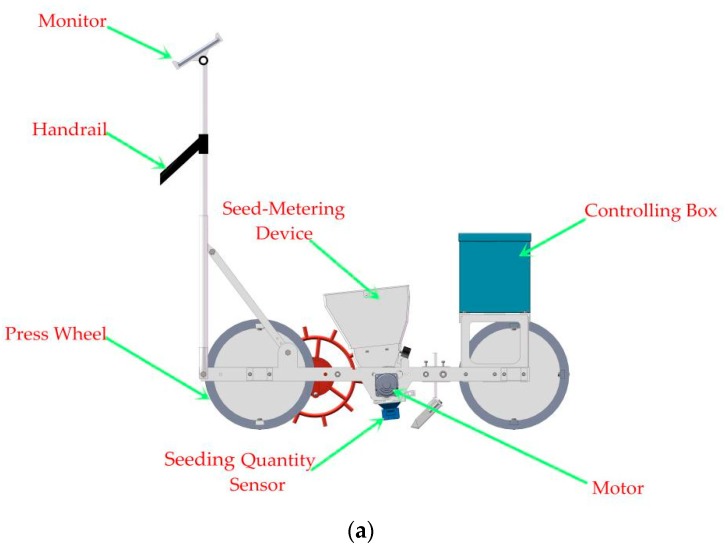
(**a**) A whole seeding unit and (**b**) a structure diagram of the seed-metering device and the seeding quantity sensor. 1. Seeding plate; 2. Hole; 3. Seed; 4. Seeding quantity sensor; 4-1. Taper inlet; 4-2. Infrared ray receiver; 4-3. Infrared ray transmitter; 4-4. Seed-guiding slot. 5. Seed-protecting device. The L and W represent the length and width of the seed-guiding slot.

**Figure 2 sensors-19-05191-f002:**
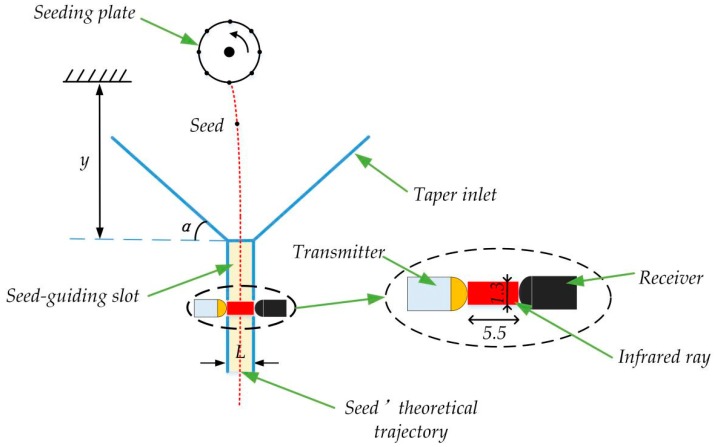
The falling trajectory when a seed falls through the seed-guiding slot. The unit of the number in this figure is mm.

**Figure 3 sensors-19-05191-f003:**
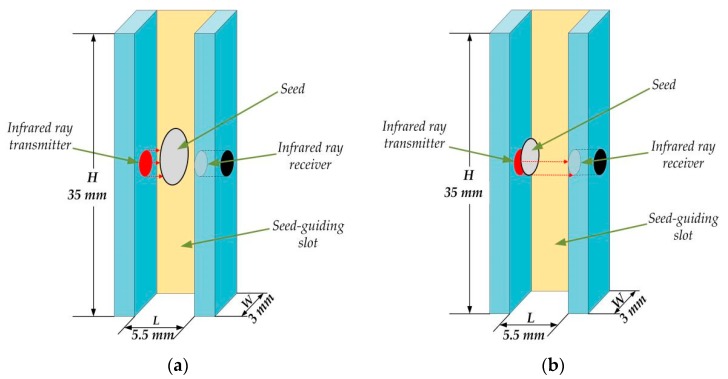
Two situations when a seed falls through the seed guiding slot: (**a**) the diameter of a relatively big seed must be less than *W* and (**b**) a small seed must shade at least two-fifths infrared ray. The *L*, *W*, and *H* in the figures represent the length, width, and height of the seed-guiding slot.

**Figure 4 sensors-19-05191-f004:**
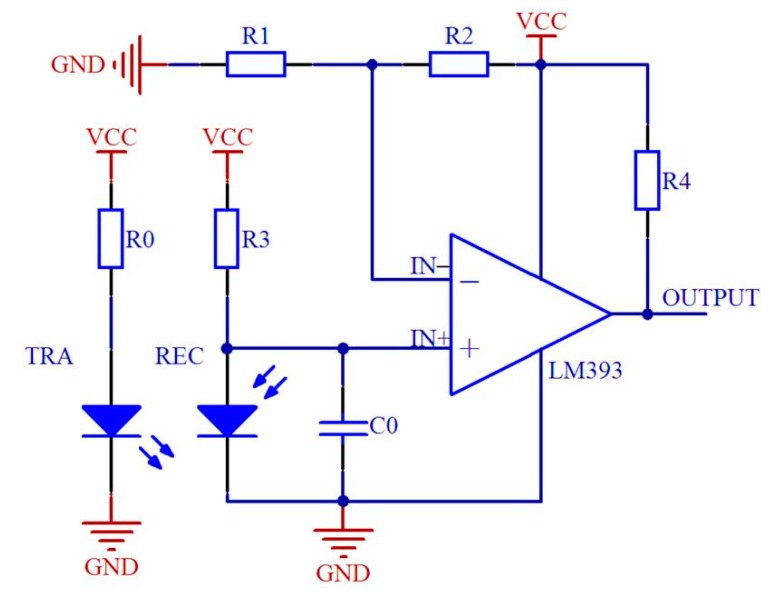
Schematic diagram of the signal processing circuit. TRA and REC refer to the infrared ray transmitter and receiver, respectively.

**Figure 5 sensors-19-05191-f005:**
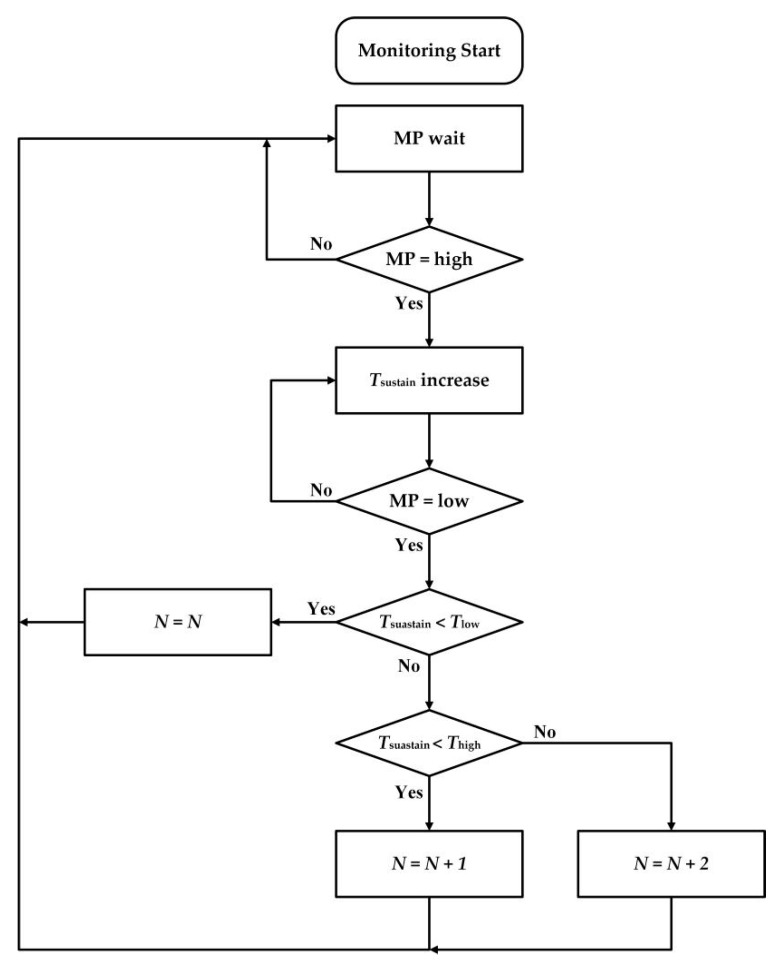
Flow chart of the SCA. MP: monitoring pin.

**Figure 6 sensors-19-05191-f006:**
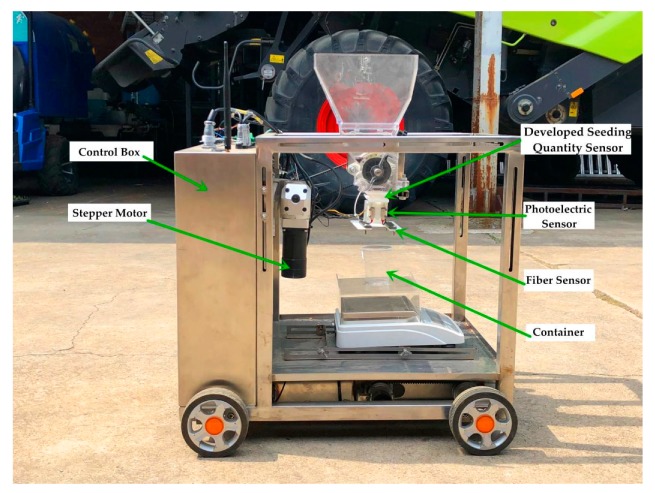
Seeding experimental apparatus.

**Figure 7 sensors-19-05191-f007:**
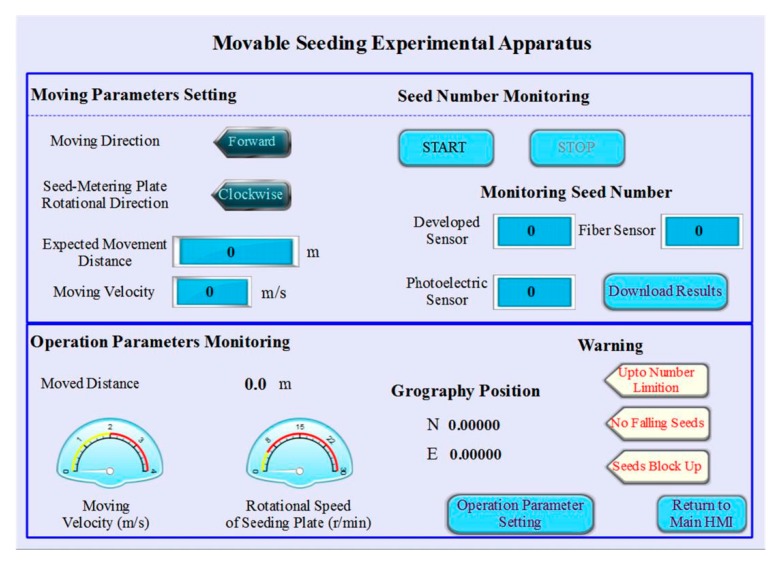
Sketch of the human–machine interface (HMI).

**Figure 8 sensors-19-05191-f008:**
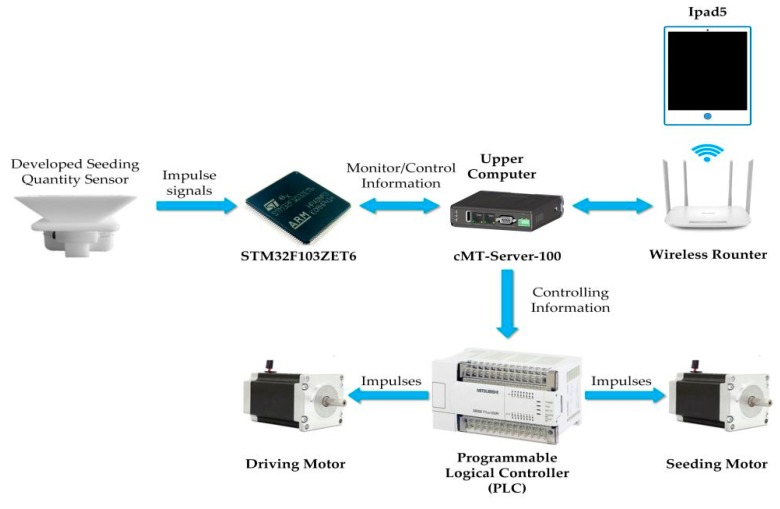
Components of the seeding experimental apparatus hardware system.

**Figure 9 sensors-19-05191-f009:**
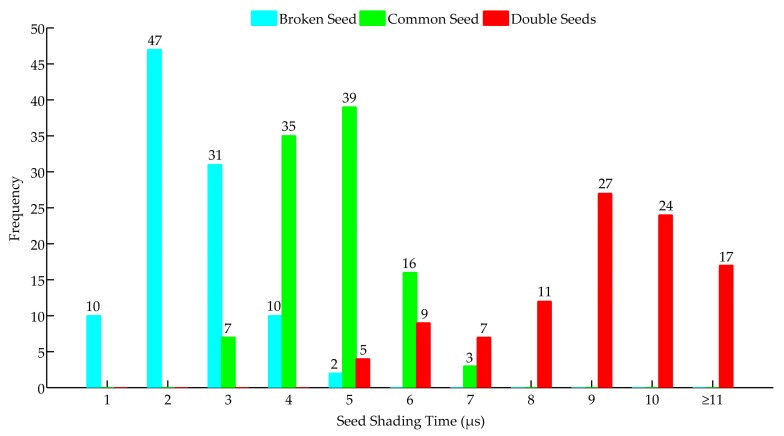
The frequency of the seed-shading time under three conditions.

**Figure 10 sensors-19-05191-f010:**
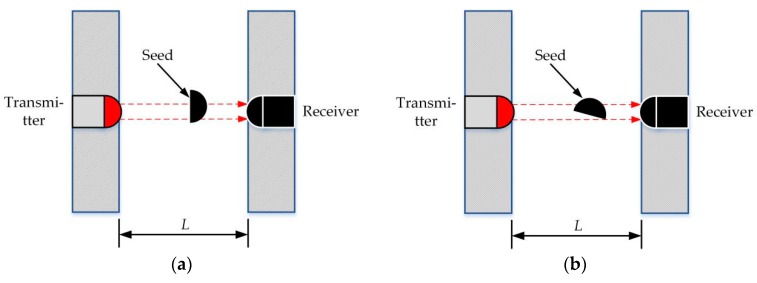
Two seed-falling postures when a broken seed falls through the seed-guiding slot: (**a**) a broken seed falls along its diameter direction and (**b**) a broken seed falls with an arbitrary posture.

**Figure 11 sensors-19-05191-f011:**
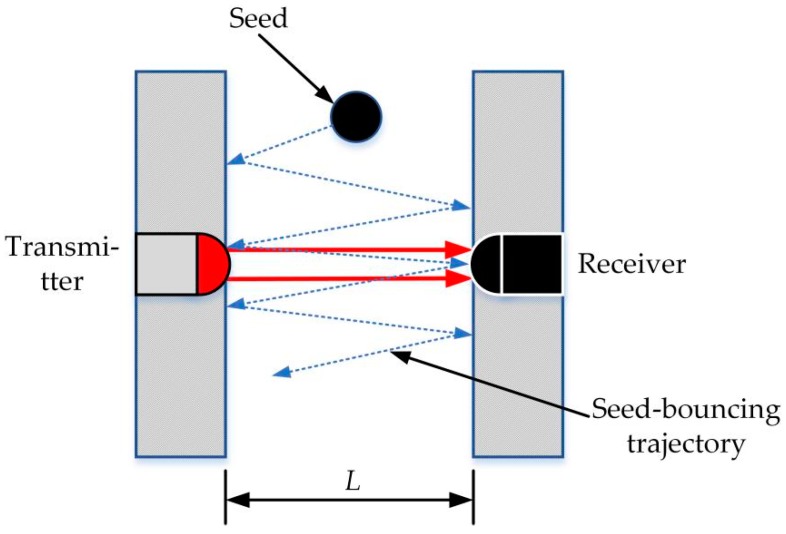
A seed might bounce between within the seed-guiding slot.

**Figure 12 sensors-19-05191-f012:**
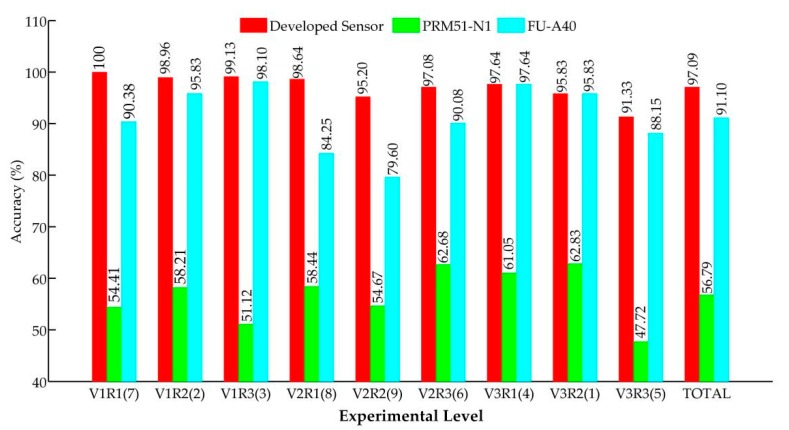
Accuracy of the developed sensor, photoelectric sensor, and fiber sensor under each experimental level. V and R represent the forward velocity of the experimental apparatus and the rotational speed of the seeding plate. The numbers 1, 2, and 3 indicate the level 1, 2, and 3 of each factor. The number in bracket is the performing sequence of each experiment.

**Table 1 sensors-19-05191-t001:** Level values of each factor of the structure optimization experiment.

Factor	Minimum	Low	Mean	High	Maximum
Slot length *L* (mm)	2.14	3	4.25	5.5	6.35
Slot width *W* (mm)	2.32	2.5	2.75	3	3.17
Inlet angle *α* (°)	19.77	30	45	60	70.22

Note: The low and high levels are the theoretical value range. The maximum and minimum are the axis points of the RSM.

**Table 2 sensors-19-05191-t002:** Factors and levels in the accuracy comparison experiment.

Factor	Level 1	Level 2	Level 3
Forward Velocity (m∙s^−1^)	0.5	1.0	1.5
Rotational speed (rev∙min^−1^)	5	12.5	20

**Table 3 sensors-19-05191-t003:** Average monitoring accuracy under each level combination.

Slot Length *L* (mm)	Slot Width *W* (mm)	Inlet Angle *α* (°)	Accuracy (%)
3	2.5	30	99.98
5.5	2.5	30	97.47
3	3	30	98.57
5.5	3	30	100.00
3	2.5	60	99.11
5.5	2.5	60	86.94
3	3	60	98.47
5.5	3	60	91.62
2.14	2.75	45	98.08
6.35	2.75	45	96.10
4.25	2.32	45	99.04
4.25	3.17	45	87.13
4.25	2.75	19.77	98.06
4.25	2.75	70	98.53
4.25	2.75	45	97.22
4.25	2.75	45	97.30
4.25	2.75	45	95.12
4.25	2.75	45	97.06
4.25	2.75	45	97.22
4.25	2.75	45	97.44

**Table 4 sensors-19-05191-t004:** Factorial analysis of the monitoring accuracies obtained by the developed sensor.

Source	Sum of Squares	Df	Mean Squared	F	Sig.
Corrected model	0.002	8	0.000	1.050	0.438
Intercept	26.059	1	26.059	92,085.718	0.000
Rotational speed	0.001	2	0.000	1.555	0.238
Velocity	0.001	2	0.000	1.657	0.219
Rotational speed × Velocity	0.001	4	0.000	0.493	0.741
Error	0.005	18	0.000		
Total	26.066	27			
Corrected total	0.007	26

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
