# Peer review of "Development and Experimental Analysis of a Seeding Quantity Sensor for the Precision Seeding of Small Seeds"

_sensors, 2019, doi:10.3390/s19235191_

Round 1

Reviewer 1 Report

See attached comments

Reviewer 2 Report

The authors took devising of precision seeding performance monitoring (PSPM) sensor as the aim of conducted tests. The task of precision seeder is to evenly distribute seeds in a row during seeding. Both maintaining equal spacing between seeds and the absence of empty/skipped holes/slots are important.

Although the device proposed by authors is an interesting solution, it does not allow to monitor the quality of precision seeding. Developed device only counts seeds sowed during precision seeding. Such sensor should therefore allow both counting seeds and checking if the requirements for even distribution of seeds in a row are met.

The authors state that they will do so in further studies. Therefore the stated aim of research was not achieved. For this reason I believe that the manuscript should be thoroughly reworded

I suggest to change the title and aim of work. For example “Development and experimental analysis of seed counting sensor for precise sowing of small seeds”.

Other comments

1.

The section “2.2. Design of the Seed Falling Trajectory controlling(SFTC)Device” raises my doubts.

If the seeds leave the seeding plate evenly and in the same position, the velocity vector should be the same (similar) for subsequent seeds. All grains falling under the influence of gravity should follow a similar trajectory and reach the lower position at the same time (seeding into the soil). If their movement was chaotic and the falling time differed then uneven placement in the soil would occur.

If the device designed by the authors allows the possibility of differences in the trajectory of seed movement, their placement in the soil will be uneven despite the fact that they will leave the seeding plate evenly. Therefore, the SFTC device is badly designed.

Analysis of grain movement and reflection in the device raises doubts - what shape do grains have?

I suggest removing the entire subsection 2.2.

2.

The "3. Materials and Methods" section should include a subsection, “Statistical calculation methodology”.

3.

In subsection “3.1. Intelligent seed number judging algorithm (ISJA)” the authors use the term intelligent algorithm. It has been accepted in the literature that the terms "intelligent algorithm" or "intelligent method" are used when artificial intelligence methods are used (artificial neural networks, fuzzy logic, genetic algorithms).

The authors in the proposed algorithm do not use any methods of artificial intelligence.

I suggest deleting the word intelligent.

4.

In the Conclusion section, the authors state: “The results demonstrated that the developed PSPM sensor is statistically steadily and robust in the mobile condition.”

The statement "mobile conditions" raises doubts.

In real conditions of precision sowing, the seeding section moving in a field is subject to greater vibrations than during the experiment carried out by the authors (concrete ground).

I suggest using the term: “in the experiment conditions”.

Round 2

Reviewer 1 Report

The authors have done a great work to improve the overall quality of the paper. I still have some doubt regarding the speed (1.5 m/s): I understand that it is quite standard in China, but the international trend is to go to higher speeds, also in horticulture. 

The second doubt is related to vibrations: even though the soil is very flat, vibration might be generated by the tractor, through the PTO. 

However consider these comments as additional optional improvements, which might further increase the quality of the paper. 

Reviewer 2 Report

The authors responded to all my comments and made significant changes to the manuscript.

I believe that in its current form the publication is suitable for publication in the Sensors journal.